# Artificial Intelligence in Risk Stratification and Outcome Prediction for Transcatheter Aortic Valve Replacement: A Systematic Review and Meta-Analysis

**DOI:** 10.3390/jpm15070302

**Published:** 2025-07-11

**Authors:** Shayan Shojaei, Asma Mousavi, Sina Kazemian, Shiva Armani, Saba Maleki, Parisa Fallahtafti, Farzin Tahmasbi Arashlow, Yasaman Daryabari, Mohammadreza Naderian, Mohamad Alkhouli, Jamal S. Rana, Mehdi Mehrani, Yaser Jenab, Kaveh Hosseini

**Affiliations:** 1Tehran Heart Center, Cardiovascular Diseases Research Institute, Tehran University of Medical Sciences, Tehran 1411713138, Iran; ssh-shojaei@student.tums.ac.ir (S.S.); a-mousavi@student.tums.ac.ir (A.M.); s-kazemian@alumnus.tums.ac.ir (S.K.); sabamlki75@gmail.com (S.M.); mehdi.mehrani@gmail.com (M.M.); yasjenab@gmail.com (Y.J.); 2School of Medicine, Tehran University of Medical Sciences, Tehran 1936893813, Iran; 3Students’ Scientific Research Center, Tehran University of Medical Sciences, Tehran 1417755331, Iran; 4Heart Failure Research Center, Cardiovascular Research Institute, Isfahan University of Medical Sciences, Isfahan 7346181746, Iran; shivaarmaniii@gmail.com; 5Medical Students Research Centre, Iran University of Medical Sciences, Tehran 1449614535, Iran; farzin.tahmasby@gmail.com; 6Pediatric Urology and Regenerative Medicine Research Center, Children’s Medical Center, Gene, Cell & Tissue Research Institute, Tehran University of Medical Sciences, Tehran 1419733151, Iran; yasaman.daryabari1395@gmail.com; 7Department of Cardiovascular Medicine, Mayo Clinic, Rochester, MN 55905, USA; naderian.mohammadreza@mayo.edu (M.N.); adnanalkhouli@gmail.com (M.A.); 8Division of Cardiovascular Diseases, Department of Medicine, West Virginia University, Morgantown, WV 26506, USA; 9Department of Cardiology, Oakland Medical Center, Kaiser Permanente Northern California, Oakland, CA 94611, USA; jamal.s.rana@kp.org

**Keywords:** transcatheter aortic valve replacement, artificial intelligence, all-cause mortality, cardiovascular event

## Abstract

**Background/Objectives:** Transcatheter aortic valve replacement (TAVR) has been introduced as an optimal treatment for patients with severe aortic stenosis, offering a minimally invasive alternative to surgical aortic valve replacement. Predicting these outcomes following TAVR is crucial. Artificial intelligence (AI) has emerged as a promising tool for improving post-TAVR outcome prediction. In this systematic review and meta-analysis, we aim to summarize the current evidence on utilizing AI in predicting post-TAVR outcomes. **Methods:** A comprehensive search was conducted to evaluate the studies focused on TAVR that applied AI methods for risk stratification. We assessed various ML algorithms, including random forests, neural networks, extreme gradient boosting, and support vector machines. Model performance metrics—recall, area under the curve (AUC), and accuracy—were collected with 95% confidence intervals (CIs). A random-effects meta-analysis was conducted to pool effect estimates. **Results:** We included 43 studies evaluating 366,269 patients (mean age 80 ± 8.25; 52.9% men) following TAVR. Meta-analyses for AI model performances demonstrated the following results: all-cause mortality (AUC = 0.78 (0.74–0.82), accuracy = 0.81 (0.69–0.89), and recall = 0.90 (0.70–0.97); permanent pacemaker implantation or new left bundle branch block (AUC = 0.75 (0.68–0.82), accuracy = 0.73 (0.59–0.84), and recall = 0.87 (0.50–0.98)); valve-related dysfunction (AUC = 0.73 (0.62–0.84), accuracy = 0.79 (0.57–0.91), and recall = 0.54 (0.26–0.80)); and major adverse cardiovascular events (AUC = 0.79 (0.67–0.92)). Subgroup analyses based on the model development approaches indicated that models incorporating baseline clinical data, imaging, and biomarker information enhanced predictive performance. **Conclusions:** AI-based risk prediction for TAVR complications has demonstrated promising performance. However, it is necessary to evaluate the efficiency of the aforementioned models in external validation datasets.

## 1. Introduction

Transcatheter aortic valve replacement (TAVR) has become a cornerstone therapy for patients with severe aortic stenosis, especially those at elevated surgical risk [1]. By employing small surgical incisions cuts and delivering the valve via catheter, this procedure is significantly less invasive and carries a lower risk for patients compared with traditional approaches, such as surgical aortic valve replacement (SAVR) [2]. Despite its widespread adoption, TAVR is associated with various complications and adverse events, including all-cause mortality, stroke, heart failure-related rehospitalization, and conduction disturbances [3]. As procedural techniques and device technologies evolve, TAVR is increasingly offered to broader patient populations, including those at low risk [4,5]. However, optimizing outcomes remains challenging due to the multifactorial nature of post-TAVR complications, which can significantly affect patients’ survival and quality of life [6].

The recent literature demonstrated that the occurrence of specific complications following TAVR significantly impacts long-term mortality and quality of life. These adverse events often exert a more profound influence on patient outcomes than the cumulative burden of pre-existing comorbidities as captured by traditional surgical risk scores [7,8]. Consequently, achieving an “event-free” TAVR procedure should be considered a primary objective to enhance and optimize clinical outcomes. Traditional surgical risk scores, such as the EuroSCORE and Society of Thoracic Surgeons (STS) score, have demonstrated limited discriminative ability in predicting mortality after TAVR [9].

In contrast, artificial intelligence (AI) and machine learning (ML) algorithms have demonstrated superior predictive capabilities in the TAVR setting [10]. These models can process complex, high-dimensional data and uncover intricate, non-linear relationships among variables. For example, AI models have achieved a pooled mean AUC of 0.79 in predicting post-TAVR mortality, significantly outperforming traditional scores [11]. The enhanced performance of AI models can be attributed to their ability to integrate diverse data types, including clinical, imaging, biomarker, and procedural variables [12]. By leveraging these multidimensional predictors, AI models can provide more individualized risk assessments, facilitating better patient selection and personalized treatment strategies [13]. However, AI usage should be guided under specific frameworks to ensure the effective and responsible integration of these innovative methods into healthcare [14].

Given the growing interest in AI-driven prediction models in the context of TAVR, a systematic review and meta-analysis is warranted to synthesize the current evidence and evaluate the performance of AI-based models across various clinical endpoints. The objective of this study is to systematically review and quantitatively assess the predictive performance of AI and ML algorithms used to forecast outcomes following TAVR.

## 2. Methods

The current study is reported in accordance with the preferred reporting items for systematic reviews and meta-analyses protocols (PRISMA) (Appendix A) [15]. The review protocol was registered on the Prospective Register of Systematic Reviews (PROSPERO; CRD42025636245). Due to analyzing data from previously published studies, we did not require formal ethical approval.

### 2.1. Search Strategy

A comprehensive literature search was conducted through the PubMed and Embase databases from inception to 24 September 2024 to identify relevant studies. The search strategy included the following keywords: [“TAVI” OR “TAVR”] AND [“AI”, “ML”]. Synonyms and equivalent terms for these keywords were also included to ensure a broad and inclusive search. Detailed search strategy for each database is provided in Appendix A. Reference lists of review articles and included studies were manually screened for additional relevant citations.

### 2.2. Eligibility Criteria and Screening

Studies were eligible for inclusion if they focused on TAVR and the use of ML algorithms for predicting post-TAVR outcomes. Exclusion criteria included abstracts, case reports, review articles, editorials, and animal studies. After removing duplications, two independent reviewers (SS and AM) screened titles and abstracts for relevance. Full texts of potentially eligible studies were assessed by the same reviewers. Any discrepancies between reviewers were resolved through mutual consensus and, if necessary, consultation with a third expert reviewer (KH).

### 2.3. Data Extraction

Key data were extracted by two independent authors (SS and AM), focusing on study characteristics (e.g., first author, publication year, country, design, inclusion and exclusion criteria, and sample size), population demographics (e.g., mean age and percentage of males), AI methodologies (e.g., algorithms used, feature selection processes, evaluation and validation method, and model development techniques), imaging data (e.g., cardiac magnetic resonance imaging (CMRI), echocardiography, computed tomography (CT), and electrocardiography (ECG)), study outcomes, as well as recall, specificity, negative predictor value, and positive predictive value. Performance metrics, such as AUC, accuracy, and recall, were systematically collected. Additional data on clinical endpoints and external validation were also recorded. To avoid including the same study population multiple times in our analysis, we selected the best-performing model (highest AUC) from each study among the various AUCs reported and machine learning algorithms employed. Furthermore, for studies that provided both internal and external validation data, priority was given to external validation. Discrepancies in data extraction were resolved through mutual consensus or consultation with a third reviewer (KH) when necessary.

### 2.4. Quality Assessment

Three reviewers (SM, FTA, and YD) evaluated the quality and risk of bias of the included studies using the prediction model risk of bias assessment tool (PROBAST), which assesses studies in four domains: participants, predictors, outcome, and analysis. Studies were rated as having a low, high, or unclear risk of bias based on these criteria [16]. Adherence to the transparent reporting of a multivariable prediction model for individual prognosis or diagnosis (TRIPOD) guidelines was also evaluated to ensure transparent reporting of model development, validation, and performance [17].

### 2.5. Outcomes

Our primary outcome was all-cause mortality, while secondary outcomes included study-defined major adverse cardiovascular events (MACE), new permanent pacemaker implantation or new left bundle branch block (LBBB), valve-related dysfunction, stroke, and heart failure-related rehospitalization.

### 2.6. Statistical Analysis

A random-effects meta-analysis was conducted to pool effect model performance metrics such as AUC, accuracy, and recall across studies, accounting for potential heterogeneity. Heterogeneity was quantified using the I^2^ statistic, with values above 75% indicating substantial heterogeneity. Sensitivity analysis using a leave-one-out method was performed to assess the robustness of pooled estimates by systematically excluding individual studies and evaluating their influence on the overall effect size.

Subgroup analyses were conducted to explore potential sources of heterogeneity and differences in model performance. Studies were stratified based on the type of features (baseline and clinical data, imaging data, biomarkers, and procedural data) utilized in their predictive models. We also evaluated our results based on the best performance reported for each ML model to better identify sources of heterogeneity. Publication bias was assessed using visual inspection of funnel plots for asymmetry and further evaluated statistically using the Egger’s test and Begg’s test, where applicable. All statistical analyses were performed using R software version 4.1.2 (The R Foundation, Vienna, Austria), utilizing the meta, metafor, and ggplot2 packages for meta-analysis. The summary effect sizes were reported with 95% confidence intervals (CI). We generated forest plots to visualize individual study estimates and the pooled effect.

## 3. Results

### 3.1. Study Selection and Characteristics

Our database search initially yielded 7287 studies and, after removing duplicates, 5131 studies were considered for title and abstract screening. Eventually, 158 full texts were evaluated by two independent reviewers, and 43 articles (366,269 patients) were included in the final analysis. The corresponding PRISMA flow chart and study selection process is outlined in Figure 1.

The included studies were published between 2017 and 2024 and provided original data on AI model-based prediction outcomes following TAVR, with 40 studies using internal validation models [10,18,19,20,21,22,23,24,25,26,27,28,29,30,31,32,33,34,35,36,37,38,39,40,41,42,43,44,45,46,47,48,49,50,51,52,53,54,55,56] and only 3 studies utilizing external validation models [57,58,59]. These studies included diverse cohorts with sample sizes ranging from 129 [48] to 117,389 [25] patients, with a mean age of 80 ± 8.25 years. The proportion of male participants varied between 36.3% [20] and 62% [44]. Most studies were retrospective studies, with four prospective cohorts [18,31,44,48] and only one post hoc analysis [39].

A wide range of machine learning algorithms were used in the included studies, whereas the best algorithm based on AUC was selected for each study to be included in the analyses. These included models utilized various algorithms, such as random forest (n = 4), extreme gradient boosting (n = 5), multilayer perceptron (n = 2), support vector machine [60], neural network (n = 3), logistic regression (n = 7), etc. These models were applied to different data sources, including baseline and clinical data, imaging data, biomarkers, procedural data, and a combination of them. The studies were conducted in various countries, including the United States (n = 16), Germany (n = 7), the Netherlands (n = 6), France (n = 2), etc. Details on the study characteristics are reported in Table 1.

### 3.2. Risk of Bias Assessment

The risk of bias of assessment based on TRIPOD guidelines showed that most studies met key reporting standards, though some (n = 4) lacked details on model calibration and validation (details are presented in Appendix A). The PROBAST risk assessment indicated that 4 studies had high concerns regarding overall applicability due to outcome applicability, while 40 studies had a low risk of bias. The detailed results are presented in Appendix A.

### 3.3. All-Cause Mortality

The pooled AUC of 26 models predicting all-cause mortality was 0.78 (95% CI: 0.74–0.82, I^2^=98.1%; Figure 2a), with individual model AUCs ranging from 0.55 to 0.97 [10,20,21,22,26,31,35,37,38,39,41,42,44,46,47,48,49,50,51,52,53,54,56,57,58,59]. The pooled accuracy of six models was 0.81 (95% CI: 0.69–0.89, I^2^ = 98.2%; Figure 3a) [20,35,37,46,50]. The pooled recall was 0.90 (95% CI: 0.70–0.97, I^2^ = 99.0%; Figure 4a), with recall values ranging from 0.33 to 1.00 (Table 2) [20,35,37,42,46,47,49,56].

The subgroup analysis demonstrated that the models trained exclusively on baseline and clinical data (n = 6) had a pooled AUC of 0.77 (95% CI: 0.69–0.85, I^2^ = 94.1%) [10,22,37,38,40,54]. The three models utilizing only imaging data showed a pooled AUC of 0.73 (95% CI: 0.71–0.74; I^2^ = 0%) [26,41,44]. The pooled AUC for the models incorporating baseline, clinical, and imaging data (n = 2) was 0.77 (95% CI: 0.69–0.84; I^2^ = 48.7%; Appendix A) [20,50], while the models integrating baseline and clinical, and biomarker data (n = 2) achieved a higher pooled AUC of 0.91 (95% CI: 0.88–0.95; I^2^ = 0.0%; Table 3) [39,48].

### 3.4. New Permanent Pacemaker Implantation or New Left Bundle Branch Block

For the prediction of pacemaker implantation need or left bundle branch block, the pooled AUC of nine models was 0.75 (95% CI: 0.68–0.82; I^2^ = 93.2%, Figure 2b), with individual AUC values ranging from 0.61 to 0.92 [19,27,28,33,34,37,40,45,46]. The pooled model accuracy was 0.73 (95% CI: 0.59–0.84; I^2^ = 99.1%, Figure 3b) [19,27,29,37,45,46] and the pooled recall was 0.87 (95% CI: 0.50–0.98; I^2^ = 99.1%, Figure 4b) (Table 2) [19,33,37,40,45,46].

The subgroup analysis indicated that the models trained on baseline, clinical, and imaging data (n = 2) had a pooled AUC of 0.77 (95% CI: 0.68–0.85; I^2^ = 90.1%, Appendix A) [27,34], while the addition of procedural data instead of imaging data (n = 3) further improved the pooled AUC to 0.75 (95% CI: 0.62–0.88; I^2^ = 73%, Appendix A) [28,29,40]. The highest AUC of 0.92 (95% CI: 0.84–1.00, Appendix A) was achieved in the model developed by Ouahidi et al., which integrated baseline, clinical, imaging, and procedural data (Table 3) [19].

### 3.5. Valve-Related Dysfunction

The combined AUC of the four models predicting valve-related dysfunction was 0.73 (95% CI: 0.62–0.84; I^2^ =96%, Figure 2c), with individual model AUCs ranging from 0.57 to 0.80 [23,33,37,46]. The overall accuracy of the three models was 0.79 (95% CI: 0.57–0.91; I^2^ =98.7%, Figure 3c) [23,37,46]. The pooled recall (n = 4) was 0.54 (95% CI: 0.26–0.80; I^2^ =99.0%), with individual recall values varying from 0.12 to 0.73 (Table 2) (Figure 4c) [23,33,37,46].

Abdelkhalek et al. used a model trained on baseline and clinical data, achieving an AUC of 0.74 (95% CI: 0.67–0.80) [33]. Another model utilized in the study by Shi et al., which relied solely on imaging data, had a higher AUC of 0.80 (95% CI: 0.73–0.80) [23]. However, the study by Gomes et al. used a model incorporating baseline, clinical, and imaging data, which had a lower AUC of 0.57 (95% CI: 0.52–0.62) (Table 3) (Appendix A) [46].

### 3.6. MACE

For predicting MACE, the combined AUC of five models was 0.79 (95% CI: 0.67–0.92; I^2^ = 89.9%, Figure 2d), with individual model AUCs ranging from 0.63 to 0.95 (Table 2) [24,32,36,43,55].

In a subgroup analysis, Stan et al. used a model trained on baseline and clinical data, achieving a high AUC of 0.92 (95% CI: 0.85–0.99) [24]. When models incorporated baseline, clinical, imaging, and biomarker data, the pooled AUC of two models improved to 0.84 (95% CI: 0.60–1; I^2^ = 91%) [32,43]. However, for models relying solely on imaging data (n = 2), the pooled AUC was lower at 0.67 (95% CI: 0.58–0.76; I^2^ = 91%) (Table 3) (Appendix A) [36,55].

### 3.7. Stroke

The pooled AUC of the three models predicting stroke was 0.73 (95%CI: 0.59–0.88; I^2^ = 97.1%), with individual model AUCs ranging from 0.60 to 0.82 (Figure 2e) (Table 2) [18,37,46].

### 3.8. Heart Failure-Related Re-Hospitalization

Pooling the three AI models for predicting heart failure [61]-related rehospitalization yielded an overall AUC of 0.70 (95% CI: 0.60–0.81; I^2^ = 83.3%, Figure 2f), with individual model AUCs ranging from 0.57 to 0.76 (Table 2) [22,25,30].

For the subgroup analysis based on baseline and clinical data, pooling two models resulted in an AUC of 0.67 (95% CI: 0.49–0.86; I^2^ = 91.2%) [22,30]. Sulaiman et al. reported a higher AUC of 0.74 (95% CI: 0.70–0.78) when integrating baseline, clinical, and procedural data (Table 3) (Appendix A) [25].

### 3.9. Sensitivity Analysis and Publication Bias

The subgroup analysis based on the type of ML algorithm regarding all-cause mortality and new PPI—the outcomes with the most included studies—also demonstrated a high I^2^. The details are demonstrated in Appendix A. The leave-one-out method also showed that removing any of the studies did not reduce the overall heterogeneity. We found no evidence of publication bias for all-cause mortality (Egger’s test *p*-value: 0.19) and new PPI or LBBB (Egger’s test *p*-value: 0.38). The funnel plots are presented in Appendix A.

## 4. Discussion

This systematic review of 43 studies, including 366,269 patients with severe AS undergoing TAVR, highlights the potential of AI models in predicting various outcomes following TAVR. Our findings showed that these models exhibit vigorous performance in predicting all-cause mortality, the need for new permanent pacemaker implantation, valve-related dysfunction, and major adverse cardiac events. Despite some concerns regarding bias and applicability in certain studies, the overall results demonstrate that integrating diverse clinical, imaging, and biomarker data can enhance predictive accuracy. Additionally, the absence of publication bias reinforces the reliability of the findings, which emphasize the promising role of AI in improving patient management and decision-making in TAVR procedures. However, the high heterogeneity observed in most of our analyses might affect the overall interpretation of our findings.

The usage of AI in predicting post-TAVR outcomes is a topic of concern which was evaluated in notable studies. Hu et al. conducted a study utilizing LR to predict post-TAVR outcomes. In this model, they demonstrated that some preoperative parameters, such as the duration of QRS in the ECG or the calcification score of the aortic valve can be predictive factors for high degree AV block after the procedure [62]. Moreover, in a study conducted by Kurmanaliyev et al., employing fine-tuned machine learning models suggested that the diameter of the left femoral artery, besides the aortic valve calcification score, was a predicting factor of early safety outcomes after TAVR. They observed that patients with lower diameter and higher calcification scores are more prone to early post-procedure adverse outcomes [63]. Whereas, in a systematic review by Sulaiman et al., it was demonstrated that various machine learning algorithms could potentially predict post-TAVR outcomes which could have been utilized in clinical settings and elevating patient-centered care [64].

The application of ML to predict patient outcomes extends beyond the realm of TAVR to a wide range of procedures. For example, in PCI, ML models have proven remarkably effective at forecasting risks like long-term all-cause mortality [65] and MACE in STEMI patients [66]. One study involving over 4500 participants demonstrated that various ML models, such as distributed random forest (DRF) and GBM, could identify high-risk STEMI patients with a high accuracy (AUCs of 0.92 and 0.91, respectively) [67]. Similarly, in patients with STEMI and diabetes, the CatBoost model outperformed traditional risk scores like GRACE, achieving an AUC of 0.92 for predicting in-hospital mortality [68]. These findings emphasize ML’s ability to refine risk assessment in time-sensitive cardiac emergencies. Regarding CABG and SAVR, algorithms like decision trees and random forests have consistently outperformed conventional methods. Decision tree models, for instance, have shown impressive accuracy in predicting short-term mortality after on-pump CABG, achieving AUCs of 0.90 and 0.86 [69]. Moreover, in patients with rheumatic heart disease undergoing valve surgery, ML models such as random forest and artificial neural networks (ANNs) have achieved perfect accuracy (AUCs of 0.98 and 0.952, respectively) in predicting in-hospital mortality [70]. Furthermore, ML’s applicability extends beyond predicting clinical outcomes. A study by Zea-Vera et al. demonstrated that extreme gradient boosting (XGBoost) algorithms can accurately predict not only operative mortality (accuracy 95%) and major morbidity/mortality (accuracy 71%), but also high hospitalization costs (accuracy 84%) across a diverse range of cardiac surgeries, including CABG, valve, and aortic procedures [71]. This reinforces the generalizability of ML across various cardiac procedures, supporting its potential as a versatile tool for risk stratification and outcome prediction in interventional cardiology. This capability to predict resource utilization offers a significant advantage for healthcare systems, enabling better planning and resource allocation.

Our systematic review and meta-analysis primarily focused on key outcomes such as mortality, MACE, PPI, hospitalization for heart failure, stroke, bundle branch block, and valve-related dysfunctions. However, the broader literature indicates that ML models are increasingly used to predict a wider range of outcomes in cardiovascular interventions, including TAVR and other procedures. This expansion reflects a more holistic approach to patient care and risk assessment, moving beyond traditional endpoints to include complications like acute kidney injury (AKI) and prolonged ventilation [72,73,74]. For instance, a study by Chong et al. used ANNs to predict reintubation and prolonged mechanical ventilation after CABG, achieving AUCs of 0.65 and 0.72, respectively [75]. Furthermore, AI models are being utilized in congenital heart surgery to predict not only mortality but also prolonged hospital or ICU stays and postoperative complications. This expanded scope is particularly relevant in complex surgical populations where traditional risk assessments may be less effective [71]. In the context of transcatheter mitral valve replacement (TMVR), ML is being explored to predict early safety outcomes, including all-cause mortality, stroke, life-threatening bleeding, AKI, coronary artery obstruction, major vascular complications, and valve-related dysfunction requiring repeat procedures. A retrospective study involving 224 participants with severe aortic stenosis found that ML models outperformed established risk scores in predicting TMVR success [63]. This capability to predict procedural outcomes represents a significant advancement beyond traditional risk stratification, which primarily focuses on adverse events. By guiding patient selection and procedural planning, ML can play a crucial role in optimizing TMVR outcomes and tailoring patient-based therapeutic plans. It was suggested that these AI-based quantification tools demonstrate superior performance to traditional previous risk scores, such as EuroSCORE and STS score, in predicting comprehensive, varied, and long-term outcomes [76]. Moreover, a fully automated prediction approach significantly reduced the time consumed per patient, which is crucial in the holistic view of clinical workflows [44]. Furthermore, physicians should be properly educated on how to use these novel methods in the most efficient and productive way, which requires dedicated training programs [77]. Notably, strict guidelines should be employed in order to prohibit the unregulated and potentially harmful use of AI technologies in ethical, legal, and professional manners [14].

In this study, we observed that ML models were employed to integrate a wide range of features including baseline clinical characteristics, imaging data, laboratory biomarkers, and procedural variables. This capacity to integrate different feature classes into a single model represents a key strength of ML over traditional risk scores, which are often limited by static variables and linear assumptions. ML models can capture complex, non-linear relationships and interactions among features, offering more nuanced risk stratification. Algorithms such as LR, SVM, and gradient boosting are particularly effective for structured data and have been widely used in predictive modeling for clinical outcomes [78]. However, these ML approaches may be insufficient for unstructured data, such as medical imaging or physiological time series. In such contexts, deep learning architectures offer a distinct advantage. For instance, convolutional neural networks (CNNs) have demonstrated efficacy in extracting morphological features from modalities like CMRI, CT, and echocardiography such as aortic valve calcification or leaflet motion directly from raw pixel data [79,80,81]. Likewise, recurrent neural networks (RNNs) are well suited to model sequential data, including intraoperative hemodynamics or ECG waveforms [82,83]. Furthermore, more advanced multimodal approaches, including ensemble learning, late fusion, and transformer-based architectures, are now being applied to combine structured and unstructured data streams, further enhancing predictive accuracy and supporting personalized decision-making [84,85,86,87,88].

The development of clinically applicable ML models for TAVR requires a comprehensive and methodologically rigorous approach. Robust performance depends not only on algorithmic design but also on access to high-quality, diverse datasets that reflect the heterogeneity of real-world TAVR populations [22]. In contrast to conventional risk scores that are derived from limited cohorts, ML models can adapt to a broader array of input features and dynamically recalibrate based on new data. Nonetheless, the clinical utility of these models extends beyond predictive accuracy. Interpretability is essential for clinician trust and uptake. Explainable AI (XAI) techniques, such as SHapley Additive exPlanations [89] and Local Interpretable Model-Agnostic Explanations (LIME), help elucidate the contribution of individual variables to model predictions, fostering transparency and clinical confidence [90]. To ensure generalizability and clinical relevance, future studies must focus on external validation in diverse populations and integration within real-world clinical workflows [91].

Future studies are needed to assess the performance of ML models in predicting the similar key outcomes we evaluated across various cardiac procedures. Additionally, further studies are required to compare the efficacy of the different ML algorithms using various metrics (e.g., AUC, recall, and calibration). Moreover, a comparison will clarify which model in specific clinical contexts was the best and the most potent. Beyond the key outcomes, studies should evaluate the utilization of ML tools for other critical outcomes, such as hospital readmission rate, cost, quality of life, and pre-procedural complications. Integrating multimodal data (e.g., imaging, biomarkers, and baseline characteristics) could further enhance predictions of outcomes. Additionally, trials with larger populations are needed to validate these models in real-world settings and assess their impact on clinical decision-making. Ultimately, detecting the most effective model of ML for each outcome and procedure might be critical for utilizing ML as a valuable predictive tool.

While AI offers promising advantages in medicine, particularly in the cardiovascular domain, there are some notable challenges that should be addressed. Incomplete medical datasets are major setbacks for the generalizability and scalability of AI. As healthcare datasets become larger and more complex, there is a need to develop effective and more efficient AI models to better perform in medical applications [92]. Additionally, the lack of a standardized framework to validate the aforementioned AI tools in real-world settings is another important issue to deal with [93]. Finally, the high computational cost and the need to improve infrastructure limits the employment of these novel methods in resource-poor settings.

Our study represents the most comprehensive systematic review and meta-analysis of data assessing the efficacy of ML models in predicting outcomes following the TAVR procedure. We conducted subgroup analyses based on multimodal data, including imaging data, clinical data, and baseline variables, which helped mitigate biases and provide more specific results. This approach aimed to identify the best-validated model for outcome prediction. However, the limited number of included studies restricted our ability to fully address the heterogeneity across all results. We also employed various approaches to address the observed heterogeneity, such as subgroup analyses stratified by ML algorithm and the leave-one-out sensitivity analyses. Despite these efforts, the overall heterogeneity remains substantial. The inability to perform additional subgroup analyses based on the type of model validation or study design, due to the limited number of studies with external validation or prospective design, likely contributed to the persistent heterogeneity. Moreover, the differences between the cut-off values among the included studies for reporting performance metrics might also be an important limitation in our analysis. Furthermore, one of the limitations of our study design is publication bias, as systematic reviews often rely on non-population-based data utilization. Finally, due to the limited data in our studies, post-treatment TAVR strategies were not adjusted in the analysis.

## 5. Conclusions

Overall, our findings emphasize the potential role of AI in patient management after TAVR. Healthcare providers can utilize these advanced predictive models to better identify high-risk patients and tailor person-centered interventions. These differences could ultimately lead to better clinical outcomes. These outcomes might potentially revolutionize routine clinical practice and cardiovascular care. 

## Figures and Tables

**Figure 1 jpm-15-00302-f001:**
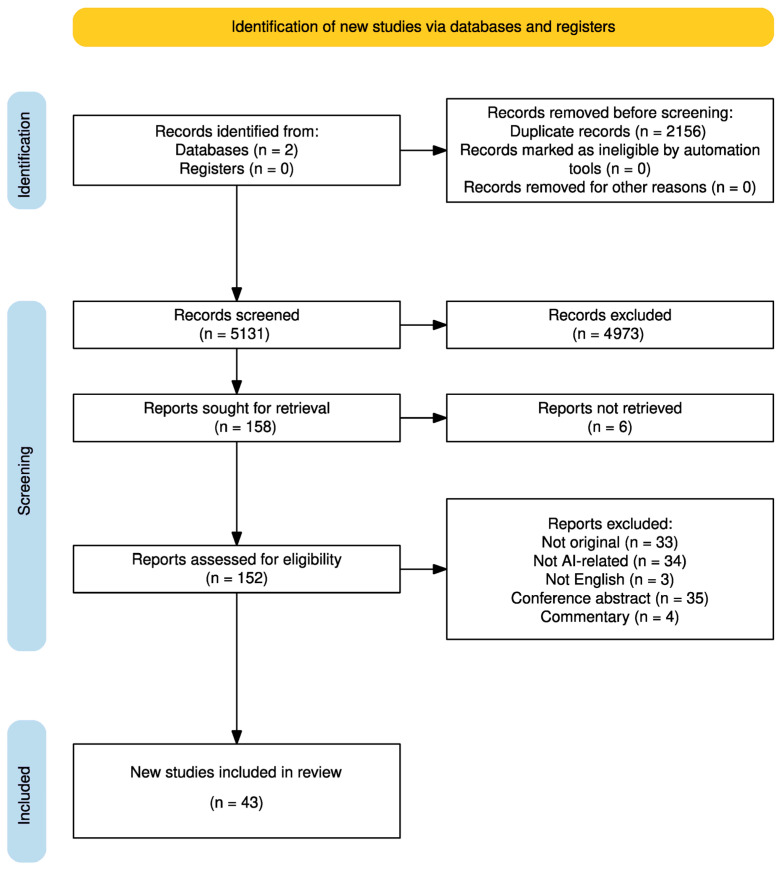
PRISMA.

**Figure 2 jpm-15-00302-f002:**
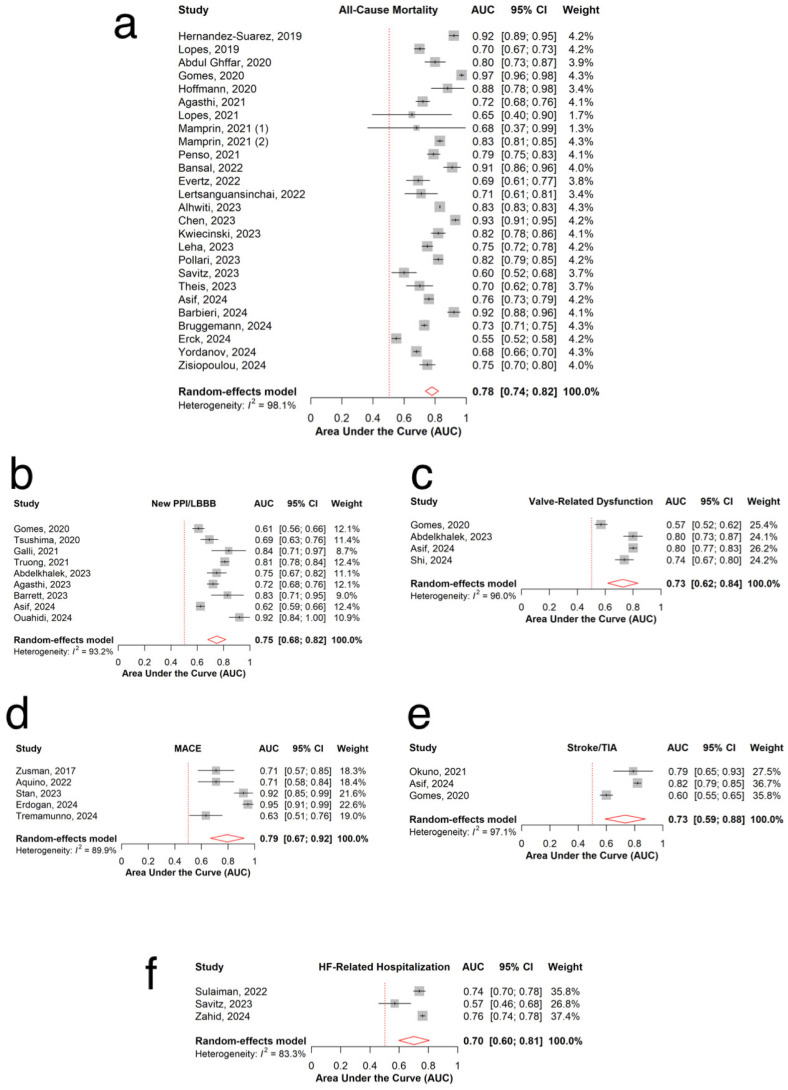
Overall AUCs for (**a**) all-cause mortality [10,20,21,22,26,31,35,37,38,39,41,42,44,46,47,48,49,50,51,52,53,54,56,57,58,59] (**b**) new PPI/LBBB [19,27,28,33,34,37,40,45,46] (**c**) valve-related dysfunction [23,33,37,46] (**d**) MACE [24,32,36,43,55] (**e**) stroke/TIA [18,37,46] (**f**) HF-related hospitalization [22,25,30].

**Figure 3 jpm-15-00302-f003:**
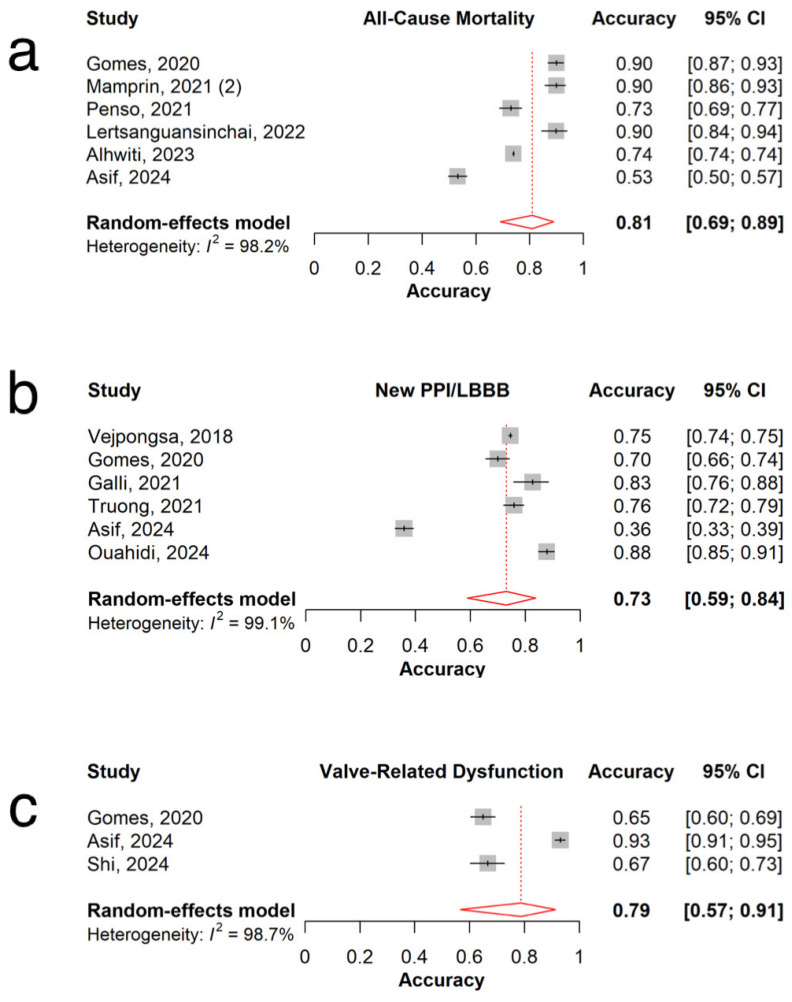
Overall accuracy for (**a**) all-cause mortality [20,35,37,46,50,53] (**b**) new PPI/LBBB [19,27,29,37,45,46] (**c**) valve-related dysfunction [23,37,46].

**Figure 4 jpm-15-00302-f004:**
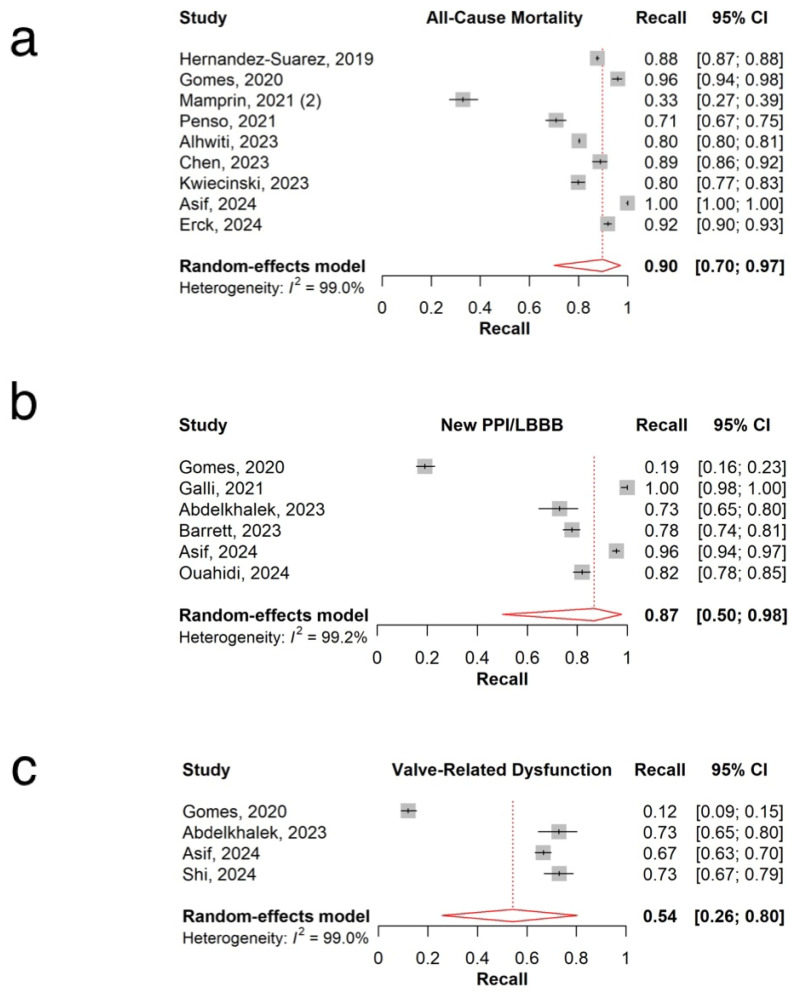
Overall recall for (**a**) all-cause mortality [20,35,37,42,46,47,49,53,56] (**b**) new PPI/LBBB [19,33,37,40,45,46] (**c**) valve-related dysfunction [23,33,37,46].

**Table 1 jpm-15-00302-t001:** Characteristics of included studies.

Author Year	Country	Study Design	Overall Dataset Size	% of Male Participants	Mean ± SD Age of Participants	Outcomes	Algorithm	Architecture	Model Development	Validation Method
Asif 2024 [37]	US	Retrospective cohort	834	N/A	N/A	Valve-related dysfunction	MLP	Initial architecture search performed using Hyperopt, a Python library built for automatic model selection and hyperparameter optimization	Baseline and clinical data	N/A
						New PPI/LBBB				
						Stroke				
						All-cause mortality				
Barbieri 2024 [39]	Austria	Post hoc analysis of a retrospective study	3079	52.19	N/A	All-cause mortality	ABC-AS score	N/A	Baseline and clinical data + biomarkers	N/A
Bruggemann 2024 [41]	Switzerland	Retrospective analysis	1449	52.42	N/A	All-cause mortality	DNN	CT images 3D deep neural network	Imaging data	Cross-validation
Erck 2024 [42]	Netherlands	Retrospective cohort	1199	47.0	80 ± 7	All-cause mortality	Adjusted model (intermuscular adipose tissue with deep learning model)	CT images	Baseline and clinical data + imaging data + procedural data	N/A
Erdogan 2024 [43]	Turkey	Retrospective cohort	453	40.8	76.1 ± 6.6	MACE	XGBoost	N/A	Baseline and clinical data + imaging data + biomarkers	N/A
Ouahidi 2024 [19]	France	Retrospective cohort	520	51.8	84.3 ± 5.5	New PPI/LBBB	SVM	N/A	Baseline and clinical data + imaging data + procedural data	Cross-validation
Shi 2024 [23]	China	Retrospective cohort	234	55.6	74.34 ± 7.62	Valve-related dysfunction	LASSO	N/A	Baseline and clinical data + imaging data	Cross-validation
Tremamunno 2024 [55]	USA	Retrospective cohort	648	58.9	77 ± 9.3	MACE	cVAE	CT images	Imaging data	N/A
Yordanov 2024 [59]	Netherlands	Retrospective cohort	16,661	50.8	79.6	All-cause mortality	Central	N/A	Baseline and clinical data + biomarkers + procedural data	Cross-validation
Zahid 2024 [30]	US	Retrospective cohort	92,363	N/A	N/A	HF-related hospitalization	LR	N/A	Baseline and clinical data	N/A
Zisiopoulou 2024 [31]	Germany	Prospective cohort	284	51.76	81.03 ± 4.75	All-cause mortality	LR	N/A	N/A	N/A
Abdelkhalek 2023 [33]	Canada	Retrospective cohort	133	57.9	81.33 ± 7.49	New PPI/LBBB	MMLR	CT images	Imaging data	N/A
						Valve-related dysfunction				
Agasthi 2023 [34]	US	Retrospective cohort	657	42.6	80.7 ± 8.2	New PPI/LBBB	GBM	N/A	Baseline and clinical data + imaging data	Cross-validation
Alhwiti 2023 [35]	US	Retrospective cohort	54,739	53.9	79.65 ± 8.5	All-cause mortality	GBM	N/A	Baseline and clinical data	Cross-validation
Barrett 2023 [40]	US	Retrospective cohort	606	N/A	N/A	New PPI/LBBB	PRIME	N/A	Baseline and clinical data + procedural data	N/A
Chen 2023 [56]	UK	Retrospective cohort	450	51.0	82.43 ± 5.21	All-cause mortality	GBST	N/A	N/A	Cross-validation
Kwiecinski 2023 [49].	Multinational	Retrospective cohort	823	46.0	82 ± 5	All-cause mortality	XGBoost	N/A	Baseline and clinical data + imaging data + biomarkers + procedural data	Cross-validation
Leha 2023 [58]	Germany	Retrospective cohort	28,147	46.8	81 ± 6.1	All-cause mortality	RF	N/A	Procedural data	Cross-validation
Pollari 2023 [21]	Germany	Retrospective cohort	629	45.0	81.9 (53.8–94.5)	All-cause mortality	Bayes	N/A	Baseline and clinical data + imaging data + biomarkers	Cross-validation
Savitz 2023 [22]	US	Retrospective cohort	1565	56.6	81 ± 8.2	HF-related hospitalization	GBM	N/A	Baseline and clinical data	Cross-validation
Stan 2023 [24]	Romania	Retrospective cohort	338	60.3	76 (71–80)	MACE	XGBoost	N/A	Baseline and clinical data	Cross-validation
Theis 2023 [26]	Germany	Retrospective cohort	760	51.0	81 ± 6	All-cause mortality	CNN	N/A	Imaging data	Cross-validation
Aquino 2022 [36]	US	Retrospective cohort	196	43.9	75 ± 11	MACE	CT-FFR with CCTA	N/A	Imaging data	N/A
Bansal 2022 [38]	US	Retrospective cohort	499	60.7	78.8 ± 9.9	All-cause mortality	RF	N/A	Baseline and clinical data	Cross-validation
Evertz 2022 [44]	Germany	Prospective cohort	142	62.0	80 (74–83)	All-cause mortality	Fully automated assessment of the volumetric parameters	Commercially available AI software provided by Neosoft (suiteHEART,)	Imaging data	N/A
Lertsanguansinchai 2022 [50]	Thailand	Retrospective cohort	178	43.8	81.6 ± 8.3	All-cause mortality	DT	N/A	Baseline and clinical data + imaging data	Cross-validation
Sulaiman 2022 [25]	US	Retrospective cohort	117,398	54.8	79.5 ± 8.4	HF-related hospitalization	LASSO	N/A	Baseline and clinical data + procedural data	Random split
Agasthi 2021 [10]	US	Retrospective cohort	1055	58.2	80.9 ± 7.9	All-cause mortality	GBM	N/A	Baseline and clinical data	Cross-validation
Galli 2021 [45]	Multinational	Retrospective cohort	151	N/A	N/A	New PPI/LBBB	K-nearest neighbors ML model	Multi-slice CT	Imaging data+ procedural data	Cross-validation
Lopes 2021 [51]	Netherlands	Retrospective cohort	1791	55.66	N/A	All-cause mortality	XGBoost	N/A	Baseline and clinical data + imaging data + biomarkers	Cross-validation
Mamprin 2021 (1) [57]	Netherlands	Inter center Cross-validation study	1931	48.05	N/A	All-cause mortality	CatBoost	N/A	Baseline and clinical data + imaging data + biomarkers	Cross-validation
Mamprin 2021 (2) [53].	Netherlands	Retrospective analysis	270	52.0	80.7 ± 6.2	All-cause mortality	CatBoost	N/A	Baseline and clinical data + imaging data + biomarkers + procedural data	Cross-validation
Okuno 2021 [18]	France	Prospective cohort	2279	52.0	83.2 years (interquartile range [IQR] 79.4–86)	MACE	Encoder–Decoder NN	N/A	Imaging data	Random split
Penso 2021 [20]	Italy	Retrospective cohort	471	36.3	81 ± 6	All-cause mortality	MLP	N/A	Baseline and clinical data + imaging data	Cross-validation
Truong 2021 [27]	US	Retrospective cohort	557	52.0	80 ± 9	New PPI/LBBB	RF	N/A	Baseline and clinical data + imaging data	Random split
Gomes 2020 [46]	Germany	Retrospective analysis	451	N/A	N/A	Valve-related dysfunction	XGBoost	N/A	Baseline and clinical data + procedural data	Cross-validation
						All-cause mortality				
						Stroke/TIA				
						New PPI/LBBB	SVM			
Abdul Ghffar 2020 [54]	US	Retrospective cohort	143	50.0	79.39 (75.07, 84.36)	All-cause mortality	N/A	N/A	Baseline and clinical data	Cross-validation
Tsushima 2020 [28]	US	Retrospective cohort	888	N/A	N/A	New PPI/LBBB	LR	N/A	Baseline and clinical data + procedural data	Random split
Hernandez-Suarez 2019 [47]	US	Retrospective cohort	10,883	52.3	81 ± 8.5	All-cause mortality	LR	N/A	Baseline and clinical data + procedural data	N/A
Hoffmann 2020 [48]	Germany	Prospective cohort	129	58.9	82.67 ± 5.25	All-cause mortality	Gradient-boosted trees (linear predictor score)	N/A	Baseline and clinical data + biomarkers	N/A
Lopes 2019 [52]	Netherlands	Retrospective analysis	1478	45.0	82.43 ± 6.23	All-cause mortality	RF	N/A	Baseline and clinical data + imaging data + biomarkers	N/A
Vejpongsa 2018 [29]	US	Retrospective cohort	18,400	N/A	N/A	New PPI/LBBB	LR	N/A	Baseline and clinical data + procedural data	N/A
Zusman 2017 [32]	Israel	Retrospective cohort	435	43.0	82.67 ± 5.21	MACE	LR	N/A	Baseline and clinical data + imaging data + biomarkers	Cross-validation

Abbreviations: MLP—Multilayer Perceptron; DNN—Deep Neural Network; SVM—Support Vector Machine; LR—Logistic Regression; LASSO—Least Absolute Shrinkage and Selection Operator; GBM—Gradient Boosting Machine; XGBoost—Extreme Gradient Boosting; RF—Random Forest; DT—Decision Tree; CNN—Convolutional Neural Network; cVAE—Conditional Variational Autoencoder; CatBoost—Categorical Boosting; SHAP—SHapley Additive exPlanations; GBST—Gradient-Boosted Survival Trees; RFC—Random Forest Classifier; I2I—Image-to-Image network; PRIME—Predictive Risk Modeling Evaluation; Encoder–Decoder NN—Encoder–Decoder Neural Network; Bayes—Bayes Classifier; MMLR—Multinominal Mutilvariate Logistic Regression; MACE—Major Adverse Cardiac Event; HF-related hospitalization—Heart Failure-related hospitalization; New PPI—New Permanent Pacemaker Implantation; LBBB—Left Bundle Branch Block; N/A—Not Applicable.

**Table 2 jpm-15-00302-t002:** Outcomes.

Outcome Category		AUC		Accuracy		Recall	
		Estimate (95% CI)	Heterogeneity (I^2^)	Estimate (95% CI)	Heterogeneity (I^2^)	Estimate (95% CI)	Heterogeneity (I^2^)
Clinical outcomes	All-Cause Mortality	0.78 (0.74, 0.82)	98.1%	0.81 (0.69, 0.89)	98.2%	0.90 (0.70, 0.97)	99%
	MACE	0.79 (0.67, 0.92)	89.9%	N/A	N/A	N/A	N/A
	Stroke/TIA	0.73 (0.59, 0.88)	97.1%	N/A	N/A	N/A	N/A
	Heart Failure-Related Hospitalization	0.7 (0.60, 0.81)	83.3%	N/A	N/A	N/A	N/A
Procedural	Pacemaker and Conduction Abnormalities	0.75 (0.68, 0.82)	93.2%	0.73 (0.59, 0.84)	99.1%	0.87 (0.50, 0.98)	99.2%
	Valve-Related Dysfunction	0.73 (0.62, 0.84)	96%	0.79 (0.57, 0.91)	98.7%	0.54 (0.26, 0.80)	99%

Abbreviations: MACE—Major Adverse Cardiac Event; AUC—Area Under Curve; CI—Confidence Interval; TIA—Transient Ischemic Attack; N/A—Not Applicable.

**Table 3 jpm-15-00302-t003:** Subgroup analysis for outcomes.

Clinical Outcomes	Subgroup	Pooled Estimate	95% CI	Heterogeneity (I^2^)	Clinical Outcomes	Subgroup	Pooled Estimate	95% CI	Heterogeneity (I^2^)
All-Cause Mortality (AUC)	Baseline and Clinical Data	0.77	(0.69, 0.85)	94.1%	Pacemaker and Conduction Abnormalities (AUC)	Baseline and Clinical Data	0.61	(0.56, 0.66)	N/A
	Imaging Data	0.73	(0.71, 0.74)	0%		Imaging Data	0.75	(0.67, 0.82)	N/A
	Procedural Data	0.75	(0.72, 0.78)	N/A		Baseline and Clinical Data + Imaging Data	0.77	(0.68, 0.85)	90.1%
	Baseline and Clinical Data + Imaging Data	0.77	(0.69, 0.84)	48.7%		Baseline and Clinical Data + Procedural Data	0.75	(0.62, 0.88)	73%
	Baseline and Clinical Data + Biomarkers	0.91	(0.88, 0.95)	0%		Imaging Data + Procedural Data	0.84	(0.71, 0.97)	N/A
	Baseline and Clinical Data + Procedural Data	0.95	(0.90, 1.00)	88.3%		Baseline and Clinical Data + Imaging Data + Procedural Data	0.92	(0.84, 1.00)	N/A
	Baseline and Clinical Data + Imaging Data + Biomarkers	0.78	(0.66, 0.89)	17.4%	Valve-Related Dysfunction	Imaging Data	0.8	(0.73, 0.87)	N/A
	Baseline and Clinical Data + Imaging Data + Procedural Data	0.55	(0.52, 0.58)	N/A		Baseline and Clinical Data + Imaging Data	0.74	(0.67, 0.80)	N/A
	Baseline and Clinical Data + Biomarkers + Procedural Data	0.68	(0.66, 0.70)	N/A		Baseline and Clinical Data + Procedural Data	0.57	(0.52, 0.62)	N/A
	Baseline and Clinical Data + Imaging Data + Biomarkers + Procedural Data	0.78	(0.70,0.87)	95.1%	Heart Failure-Related Hospitalization (AUC)	Baseline and Clinical Data	0.67	(0.49, 0.86)	91.2%
MACE (AUC)	Imaging Data	0.67	(0.58, 0.76)	0%		Baseline and Clinical Data + Procedural Data	0.74	(0.70, 0.78)	
	Baseline and Clinical Data + Imaging Data + Biomarkers	0.84	(0.60, 1.00)	91%					
	Baseline and Clinical Data	0.92	(0.85, 0.99)	N/A					

MACE—Major Adverse Cardiac Event; AUC—Area Under Curve; CI—Confidence Interval; TIA—Transient Ischemic Attack; N/A—Not Applicable.

## Data Availability

The data supporting the results of this study are available in each included study.

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
