# Peer review of "Artificial Intelligence in Risk Stratification and Outcome Prediction for Transcatheter Aortic Valve Replacement: A Systematic Review and Meta-Analysis"

_jpm, 2025, doi:10.3390/jpm15070302_

Round 1
Reviewer 1 Report
Comments and Suggestions for Authors
The authors submitted a well-written manuscript for review. The analysis is comprehensive, and the inclusion of a large patient dataset significantly increases the strength and generalizability of the findings. The integration of AI methodology with outcome prediction for TAVR is timely and valuable.
However, I would like to raise a few issues for consideration that could further improve the quality and clinical utility of the manuscript.
First, a minor technical note: the current layout of Table 1 makes it difficult to read. The first few columns (author, year, country, study design) are too narrow, making it difficult to extract basic information at a glance. I suggest adjusting the format of the table to provide better visual balance between the data columns and facilitate comparison across studies. You could also rotate it 90 degrees, as with the subsequent tables.
The substantive comments concern the assessment of factors themselves. While I believe the article provides a solid overview of AI-based predictive models, several important clinical variables that are highly relevant to TAVR populations were not adequately considered or discussed:
Atrial fibrillation is common in elderly patients undergoing TAVR and is a known risk factor for thromboembolic events. It is also strongly associated with increased long-term mortality. Furthermore, the presence of AF typically requires long-term anticoagulation.
Oral anticoagulation independently increases the risk of bleeding, especially when combined with dual antiplatelet therapy, which is not uncommon in TAVR patients with concomitant coronary artery disease.
Both conditions are very common in the typical TAVR population and can significantly impact procedural safety and clinical outcomes. Including them in the discussion—especially in the context of model development and variable selection—would increase the clinical value of the work.
In addition to the above, several other real-world predictors are increasingly being recognized as influencing outcomes after TAVR, but are often underrepresented in data-driven AI models. These include frailty—Biological age and functional capacity (e.g., walking speed, grip strength) are critical prognostic factors, particularly for mortality and readmission risk. Despite the difficulty in quantifying frailty, it remains essential for decision-making and outcome stratification. Similarly, advanced comorbidities—chronic liver disease, active or recently treated malignancy, and hematologic abnormalities (e.g., thrombocytopenia, anemia) can increase the risk of bleeding or complications and are not always captured in structured data. Incorporating or at least recognizing these variables would improve the clinical relevance and practical utility of AI-based predictive strategies in TAVR. If this is not possible, I believe it would be appropriate to include a reference to these factors in the discussion or to discuss in advance the rationale for the choices made during the planning of the work. However, I encourage the authors to consider expanding the discussion to include the clinical variables listed above. Not only are they important independent predictors of outcome, but they are also very common in the typical TAVR population. Addressing them would enhance the translational value of this work and its integration into clinical practice.
Author Response
Thank you for taking the time to read our manuscript and for your insightful comments. We greatly appreciate your thorough review, as your suggestions were not only relevant but also instrumental in enhancing the quality of our work. We have carefully considered each of your points and made the necessary revisions to improve our manuscript. Your feedback has significantly contributed to our research, and we are grateful for your valuable input.
Comment a:
First, a minor technical note: the current layout of Table 1 makes it difficult to read. The first few columns (author, year, country, study design) are too narrow, making it difficult to extract basic information at a glance. I suggest adjusting the format of the table to provide better visual balance between the data columns and facilitate comparison across studies. You could also rotate it 90 degrees, as with the subsequent tables.
Response:
Thank you for your thoughtful review. To improve the readability of Table 1, we removed the “year” column, as it duplicated information already presented in the “author year” column. Additionally, we increased the table width to better display the columns. We appreciate your constructive feedback.
Comment b:
Atrial fibrillation is common in elderly patients undergoing TAVR and is a known risk factor for thromboembolic events. It is also strongly associated with increased long-term mortality. Furthermore, the presence of AF typically requires long-term anticoagulation.
Oral anticoagulation independently increases the risk of bleeding, especially when combined with dual antiplatelet therapy, which is not uncommon in TAVR patients with concomitant coronary artery disease.
Both conditions are very common in the typical TAVR population and can significantly impact procedural safety and clinical outcomes. Including them in the discussion—especially in the context of model development and variable selection—would increase the clinical value of the work. In addition to the above, several other real-world predictors are increasingly being recognized as influencing outcomes after TAVR, but are often underrepresented in data-driven AI models. These include frailty—Biological age and functional capacity (e.g., walking speed, grip strength) are critical prognostic factors, particularly for mortality and readmission risk. Despite the difficulty in quantifying frailty, it remains essential for decision-making and outcome stratification. Similarly, advanced comorbidities—chronic liver disease, active or recently treated malignancy, and hematologic abnormalities (e.g., thrombocytopenia, anemia) can increase the risk of bleeding or complications and are not always captured in structured data. Incorporating or at least recognizing these variables would improve the clinical relevance and practical utility of AI-based predictive strategies in TAVR. If this is not possible, I believe it would be appropriate to include a reference to these factors in the discussion or to discuss in advance the rationale for the choices made during the planning of the work. However, I encourage the authors to consider expanding the discussion to include the clinical variables listed above. Not only are they important independent predictors of outcome, but they are also very common in the typical TAVR population. Addressing them would enhance the translational value of this work and its integration into clinical practice.
Response:
Thank you for your valuable comments regarding potential confounding factors in our manuscript. We recognize the importance of these parameters and, during data extraction, we attempted to collect information on relevant comorbidities as thoroughly as possible. However, due to insufficient data, we were unable to perform adjusted analyses based on these factors. In response to your insightful suggestions, we have expanded the discussion to emphasize the significance of these factors in patient evaluation and to highlight the need for more comprehensive assessment of these variables in future studies. Your recommendations clearly enhance the impact of our manuscript.
Reviewer 2 Report
Comments and Suggestions for Authors
The study examines the application of artificial intelligence (AI) and machine learning (ML) for predicting outcomes in patients undergoing transcatheter aortic valve replacement (TAVR). This is a highly topical and rapidly evolving field. The integration of AI tools in TAVR risk stratification remains underrepresented in meta-analytic literature, making the manuscript an original contribution. The systematic review of 43 studies encompassing over 366,000 patients provides substantial breadth. Although original in its scope, the study largely reiterates known predictive capabilities of AI without offering novel methodological advances or new frameworks for clinical translation.
High heterogeneity (I² > 90% in many pooled estimates) is a major limitation and may affect the validity of pooled results. Only three studies used external validation, limiting the generalizability of the AI models. The selection of the best-performing AUC from each study could introduce selection bias, potentially inflating predictive performance. I suggest to:
The authors should:
-
Address the implications of high heterogeneity more critically.
-
Reconsider the method of selecting the best AUCs per study to avoid bias.
-
Improve English clarity and sentence structure.
-
Strengthen the discussion on clinical applicability and integration of AI models.
Author Response
We sincerely appreciate your detailed review of our manuscript. Thank you for dedicating your time to provide such thoughtful comments. Your observations were spot-on and have helped us identify areas for improvement. We have addressed all of your suggestions in our revisions, which we believe have strengthened the overall quality of our paper. Your constructive feedback has been invaluable to us. The detailed response is attached.

Reviewer 3 Report
Comments and Suggestions for Authors
Review for the manuscript
Journal: JPM (ISSN 2075-4426)
Manuscript ID: JPM-3664469
Type: Systematic Review
Title: Artificial intelligence in risk stratification and outcome prediction for transcatheter aortic valve replacement: A meta-analysis
Section:
Methodology, Drug and Device Discovery
Special Issue:
Artificial Intelligence in Diagnosis, Treatment and Prognosis of Cardiovascular Diseases
the efficiency of the aforementioned models in external validation datasets.
Dear Editor,
Thank you for the invitation to review for JPM. I have some comments and suggestions regarding this manuscript before it can be accepted for publication.
OVERALL COMMENTS
In this study, based on the statement that transcatheter aortic valve replacement is an optimal treatment for patients with severe aortic stenosis, offering a minimally invasive alternative to surgical aortic valve replacement, theauthors intended to perform a systematic review and meta-analysis to summarize the current evidence on utilizing Artifiial Inteligence in predicting post- transcatheter aortic valve replacement outcomes. They included 43 studies evaluating 366,269 patients and proposed that Artifical Inteligence-based risk prediction for transcatheter aortic valve replacement complications demonstrated promising performance.
TITLE
The title is “Artificial intelligence in risk stratification and outcome prediction for transcatheter aortic valve replacement: A meta-analysis”.
“I suggest: Artificial intelligence in risk stratification and outcome prediction for transcatheter aortic valve replacement: A systematic review and meta-analysis”
ABSTRACT
This section is fine. I just suggest that the authors review some grammar mistakes in this section along with the entire manuscript.
KEYWORDS
The Keywords are: Transcatheter aortic valve replacement; artificial intelligence; outcome prediction; meta-analysis
I suggest: Transcatheter aortic valve replacement; artificial intelligence; all-cause mortality cardiovascular event
INTRODUCTION
This section is adequate. I suggest including more references published in 2025.
Also in the Introduction, the authors state that "Transcatheter aortic valve replacement (TAVR) has become a cornerstone therapy for patients with severe aortic stenosis, especially those at elevated surgical risk". I suggest that they include the explanation regarding this procedure, including a little more detail and comparing the importance of this procedure with other similar.
In addition, check the way of citing references; in MDPI, references are cited in square brackets and not in parentheses. The same problem is found in lines 53-58. Please include two or more references.
If possible, please include a brief discussion of the unregulated use of AI in clinical practice
METHODS
This section was well described.
Would it be interesting to include a sensitivity analysis for different types of AI?
RESULTS
In Table 1, the first and second columns are redundant. I suggest leaving only one column with the author and year data. However, there is a need to place the references in the MDPI format, so these references will have to be included in numbers between square brackets.
Is there a reason that Asif 2024 is highligheted in yellow?
Figure 2: reduce the size of the letters “a” to “e”. The same for Figures 3 and 4.
Would it be possible to better standardize the included outcomes? Is this a limitation for the study, as well as the high heterogeneity? Pleae comment at the end of the Discussion section when we can read that “Additionally, due to the limited number of studies reporting relevant outcomes, we could not derive more detailed insights. More over, the differences between cut-off values among included studies to report performance metrics might also be an important limitation in our analysis. Furthermore, one of the limitations of our study design is publication bias, as systematic reviews often rely on non-population-based data utilization. Finally, due to the limited data in our studies, post-treatment TAVR strategies were not adjusted in the analysis.”
DISCUSSION
This section is long but is fine. Some comments:
Since there is much discussed, is there a need to keep the discussion regarding congenital heart surgery?
Please include a critical discussion regarding AI in the clinical practice and regulatory ways.
CONCLUSION
The conclusion is short but is fine.
FUTURE PERSPECTIVES
I suggest including a separate section mentioning the Future Perspectives;
What are the current and next challenges.
REFERENCES
As mentioned above, I suggest the inclusion of more references in the Introduction section (published in 2025).
Author Response
Thank you very much for your careful reading of our manuscript and for your constructive comments. We are grateful for the time and effort you invested in reviewing our work. Your feedback was highly relevant and has played a crucial role in guiding our revisions. We tried to implement all of your suggestions and believe that these changes have significantly enhanced the clarity and impact of our research. We truly appreciate your support.
Comment 1:
The title is “Artificial intelligence in risk stratification and outcome prediction for transcatheter aortic valve replacement: A meta-analysis”.
“I suggest: Artificial intelligence in risk stratification and outcome prediction for transcatheter aortic valve replacement: A systematic review and meta-analysis”
Response:
We appreciate your valuable feedback regarding the title of our manuscript. We have revised the title in accordance with your recommendations.
Comment 2:
ABSTRACT
This section is fine. I just suggest that the authors review some grammar mistakes in this section along with the entire manuscript.
Response:
Thank you for your comment. We thoroughly review the abstract section to correct the grammatical errors. We appreciate your attention to details.
Comment 3:
The Keywords are: Transcatheter aortic valve replacement; artificial intelligence; outcome prediction; meta-analysis, I suggest: Transcatheter aortic valve replacement; artificial intelligence; all-cause mortality cardiovascular event
Response:
Thank you for your valuable feedback. We changed the Keywords section based on your opinion.
Comment 4:
INTRODUCTION
This section is adequate. I suggest including more references published in 2025.
Also in the Introduction, the authors state that "Transcatheter aortic valve replacement (TAVR) has become a cornerstone therapy for patients with severe aortic stenosis, especially those at elevated surgical risk". I suggest that they include the explanation regarding this procedure, including a little more detail and comparing the importance of this procedure with other similar.
In addition, check the way of citing references; in MDPI, references are cited in square brackets and not in parentheses. The same problem is found in lines 53-58. Please include two or more references.
If possible, please include a brief discussion of the unregulated use of AI in clinical practice
Response:
Thank you for your helpful comment on the introduction section. We have incorporated more up-to-date references and added explanations of the TAVR procedure, including comparisons with similar interventions. Additionally, we provided further details on the importance of regulated AI usage in clinical medicine. All references are now cited consistently throughout the manuscript. We appreciate your attention and comprehensive review.
Comment 5:
METHODS
This section was well described.
Would it be interesting to include a sensitivity analysis for different types of AI?
Response:
Thank you for your valuable suggestion to include a sensitivity analysis for different AI methods. We conducted a subgroup analysis stratified by machine learning algorithm type for all-cause mortality and new PPI outcomes, which included the most studies. The detailed results are presented in Supplementary Figures 6–7, and the relevant information has been added to the Methods and Results sections. This analysis revealed that heterogeneity remains high even within the same ML type, indicating that this factor does not account for the observed heterogeneity.
Comment 6:
RESULTS
In Table 1, the first and second columns are redundant. I suggest leaving only one column with the author and year data. However, there is a need to place the references in the MDPI format, so these references will have to be included in numbers between square brackets.
Is there a reason that Asif 2024 is highligheted in yellow?
Response:
Thank you for your thorough review. You are absolutely correct regarding the redundancy of the first and second columns; therefore, we have removed the second column. Additionally, we have cited each article in the table to better reference the sources. Upon evaluation, we found no highlighting in the table, and there was no specific reason for its absence.
Comment 7:
Would it be possible to better standardize the included outcomes? Is this a limitation for the study, as well as the high heterogeneity? Please comment at the end of the Discussion section when we can read that “Additionally, due to the limited number of studies reporting relevant outcomes, we could not derive more detailed insights. More over, the differences between cut-off values among included studies to report performance metrics might also be an important limitation in our analysis. Furthermore, one of the limitations of our study design is publication bias, as systematic reviews often rely on non-population-based data utilization. Finally, due to the limited data in our studies, post-treatment TAVR strategies were not adjusted in the analysis.”
Response:
Thank you for your valuable comment. In our included studies, we aimed to comprehensively assess adverse outcomes following TAVR to provide a clearer perspective on the use of AI in predicting these outcomes. However, we acknowledge that the high heterogeneity may affect our analysis, a limitation we have explicitly mentioned. We have expanded our discussion of this important issue in the Discussion section.
Comment 8:
Since there is much discussed, is there a need to keep the discussion regarding congenital heart surgery?
Please include a critical discussion regarding AI in the clinical practice and regulatory ways.
Response:
Thank you for your insightful feedback. To focus on the most relevant findings, we have removed sentences related to congenital heart surgery from our discussion. Regarding other cardiac procedures, such as CABG and PCI—which is considered an alternative procedure for patients with severe stenosis—we have retained the discussion of similar studies evaluating the effectiveness of AI in these procedures to better demonstrate the generalizability of AI use in cardiovascular interventions. We have also added further explanations to our previous statements concerning the clinical application and regulatory pathways for AI in medicine. Your comments have greatly enhanced the clarity of our manuscript.
Comment 9:
I suggest including a separate section mentioning the Future Perspectives;
What are the current and next challenges.
Response:
Based on your valuable recommendations, we have added a separate section discussing the current and future challenges of AI utility in medicine, which is essential for a better understanding of the aims of our manuscript.
Round 2
Reviewer 1 Report
Comments and Suggestions for Authors
the authors have modified the manuscript, I believe it can now be considered for publication
Reviewer 2 Report
Comments and Suggestions for Authors
Despite the improvements, the manuscript still contains sporadic typographical errors (e.g., duplicated email addresses in the correspondence section, some formatting inconsistencies with inline citations). These minor issues should be addressed in the final proofreading stage.